# A Case of Hypokalemia Possibly Induced by Nafcillin

**DOI:** 10.3390/antibiotics7040108

**Published:** 2018-12-12

**Authors:** Fernando Casado, Sitarama Arvind Mudunuru, Rabih Nasr

**Affiliations:** 1Department of Internal Medicine, BronxCare Hospital Center, New York, NY 10457, USA; fecc1901@gmail.com; 2Division of Nephrology, Department of Internal Medicine, BronxCare Hospital Center, New York, NY 10457, USA; Rnasr@bronxleb.org

**Keywords:** nafcillin, hypokalemia, side effect, *Staphylococcus aureus*

## Abstract

(1) Background: The use of intravenous antibiotics for severe infections is a common practice, either as inpatient or outpatient treatment. In the case of methicillin-susceptible *Staphylococcus aureus* (MSSA), nafcillin is a commonly prescribed intravenous antibiotic, given its known efficacy to treat infections related to this organism effectively. However, it is not without side effects. (2) Methods: We present an interesting case of persistent hypokalemia in a patient after he was started on nafcillin infusion for an MSSA infection, which eventually resolved with the completion of the treatment. (3) Results: Hypokalemia is a known side effect of nafcillin infusion, and it is believed to be mainly due to its antibiotic effect as a non-absorbable ion in the distal tubule and/or intracellular redistribution due to volume depletion. (4) Conclusions: A review of the available literature revealed that hypokalemia is a known side effect of nafcillin infusion; however, if present, it is usually mild, and only a few cases of severe hypokalemia have been reported. Usually, hypokalemia resolves when the nafcillin infusion is stopped; however, in certain cases, when this is not possible, oral potassium replacement can be used while the patient is receiving nafcillin. Clinicians should be aware of this rare, but possible, complication when using nafcillin.

## 1. Introduction

Antibiotics can cause several electrolyte imbalances, including hypernatremia, hyponatremia, hyperkalemia and hypokalemia [1]. Hypokalemia is a frequent finding in the clinical setting, estimated to occur in about 3.5% of hospitalized patients. The most common causes of hypokalemia are diuretic use and gastrointestinal losses. Less common causes are renal tubular acidosis, diabetic ketoacidosis, excess insulin, primary hyperaldosteronism, ectopic adrenocorticotropic hormone production, and medications [2,3,4,5,6]. Drug-induced hypokalemia rarely occurs after starting therapy with penicillin-type antimicrobials, such as nafcillin [1,3,6,7,8].

## 2. Case Presentation

A middle-aged African male with a medical history of hypertension presented to the emergency room complaining of right-sided weakness and bilateral shoulder pain at 1 day after onset. Home medication was ibuprofen 400 mg as needed for occasional pain, and he reported using 4–5 pills a month. A review of systems was negative for fever, chills, shortness of breath, dizziness, chest pain or palpitations; there was no nausea, vomiting, diarrhea or constipation. Vital signs were normal. Physical examination showed decreased muscle power in both of the lower extremities, more prominent in the right lower limb than the left, and the strength of the upper extremities was preserved; findings that did not fully correlate with the patient’s symptoms.

Initial laboratory tests showed a pH of 7.43 and the following micronutrient blood concentrations: sodium 133 milliequivalents per litre (mEq/L), potassium 4.0 mEq/L, chloride 91 mEq/L, bicarbonate 23 mEq/L, glucose 119 milligrams per deciliter (mg/dL), blood urea nitrogen (BUN) 20 mg/dL, and serum creatinine 0.7 mg/dL. Blood parameters were as follows: hemoglobin 13.6 g/dL, hematocrit 41.6%, platelets 150,000/mm^3^, and white blood cell count 12,000/mm^3^. An electrocardiogram performed at the time of admission was normal. A magnetic resonance imaging (MRI) scan of the cervical spine revealed an anterior epidural mass causing severe central stenosis from the mid C5 level and extending through the mid C7 level, consistent with an epidural abscess. The patient underwent emergent cord decompression and abscess drainage.

The patient was started on vancomycin, ceftriaxone, and metronidazole. Blood and abscess cultures showed methicillin-susceptible *Staphylococcus aureus*, and treatment with nafcillin was started. Other medications that were administered regularly to the patient were ranitidine 150 mg two times per day and subcutaneous heparin at 5000 units, for gastric and deep venous thrombosis prophylaxis, respectively. He also received morphine as needed for pain. The patient’s hospital course was complicated by an additional bacteremia caused by the extended spectrum beta lactamase producer and carbapemenase producing *Klebsiella pneumoniae*, for which he received a short course of polymyxin and meropenem. During his hospital course, he maintained his regular frequency of bowel movements. His baseline potassium level was around 4.3 mEq/L. As noted in Table 1 and Figure 1, the patient had a period of persistent hypokalemia after nafcillin was initiated and the potassium levels remained low despite repeated intravenous replacement of potassium chloride (80 mEq per day). Urine electrolytes showed the following concentrations: potassium 63.9 mEq/L, calcium 41 mg/dL, and magnesium 102 mg/dL. Subsequently, nephrology was consulted, and it was hypothesized that his hypokalemia was possibly induced by nafcillin. The patient was receiving intravenous potassium and magnesium supplements daily based on the electrolyte levels and later switched to oral potassium and magnesium supplements prior to discharge from the hospital, as shown in Figure 1. After he completed his treatment course of nafcillin, potassium supplements were discontinued and his potassium levels returned to baseline.

## 3. Discussion

Hypokalemia is a common clinical problem, and it has several causes, including inadequate potassium intake, increased potassium entry into the cells, increased gastrointestinal losses, increased sweat losses, and increased urinary losses [6]. The cause of hypokalemia can usually be determined from an individual’s history (such as diarrhea, vomiting, or the use of diuretics); however, in some cases, the diagnosis is not readily apparent. There are two major components in the diagnostic evaluation for hypokalemia, as follows: (1) the assessment of urinary potassium excretion to distinguish renal potassium losses from other causes of hypokalemia; and (2) an assessment of the acid–base status [5]. Several medications can cause urinary potassium wasting, with diuretics being the most common cause [5,6].

Antibiotics can cause several electrolyte disturbances, including hyponatremia (such as trimethoprim and ciprofloxacin), hypernatremia (such as amphotericin B and demeclocycline), hyperkalemia (such as trimethoprim, penicillin and amphotericim B) and hypokalemia (such as amphotericin B, penicillin, aminoglycosides and capreomycin) [1]. In the case of electrolyte imbalances associated with antibiotic use, several mechanisms have been described [1,5,6].

The reviewed literature suggested that drug-induced hypokalemia rarely occurs after starting therapy with penicillin-type antimicrobials, such as nafcillin [1,3,6,7,8], with the baseline potassium level being a key determinant in the development of hypokalemia [7]. Patients with a higher baseline potassium level had a lesser likelihood of developing severe hypokalemia but had a higher likelihood of having an acute drop in the potassium level [7]. Although this patient’s baseline potassium was around 4.3 meq/L, there was an acute drop in his potassium levels to less than 3.5 meq/L once nafcillin was started. With nafcillin, induced hypokalemia is related to non-absorbable ion effects in the distal tubule and/or intracellular redistribution due to volume depletion [6,7,8]. Sodium is presented to the distal nephron with a relatively large quantity of non-absorbable anions, which leads to increased sodium reabsorption. The negative electrical gradient created by sodium reabsorption in the lumen of the cortical collecting tubule will be partially attenuated by chloride reabsorption; however, sodium is reabsorbed in exchange for potassium, leading to a potentially marked increase in potassium excretion and, hypokalemia [3,4,7].

Another drug received by our patient during his hospitalization was meropenem, which has been rarely associated with hypokalemia, which usually resolves within 2–3 days after stopping this antibiotic [9]. The mechanism is similar to that attributed to other beta lactams [9,10]. Although this could be a potential confounder in our case, it is notable that the hypokalemia in our patient was already ongoing at the time of starting meropenem, which would go against meropenem as the only contributor to our patient’s hypokalemia. The Naranjo Scale, a tool designed to assess for adverse drug reactions [9], was 6 in our case, which indicates a probable adverse drug reaction. 

When nafcillin-associated hypokalemia is identified, aggressive potassium replacement is indicated to correct the hypokalemia [8]. The problem may also be corrected by merely decreasing the dose of nafcillin [8,11]. However, if a higher dose of the antibiotic is indicated, a potassium-sparing agent (such as spironolactone or eplerenone) may be concurrently used [8], but this was not initiated in this patient because his hypokalemia improved with potassium supplements. Additionally, hypokalemia can be treated with oral potassium supplements in mild to moderate hypokalemia as it is the safest and easiest method if the patient can tolerate oral supplementation [10]. In our case, due to an aggressive infection, nafcillin was continued and a decision to provide an oral potassium supplement was made. After treatment with nafcillin was completed, oral potassium supplementation was stopped but his potassium levels remained within normal limits.

## 4. Conclusions

Hypokalemia is a complication of therapy with high-dose penicillin G and carbenicillin. Other classes of penicillin (such as nafcillin) rarely reduce serum potassium concentrations, and previous published cases mainly described mild hypokalemia with only one case involving serious hypokalemia while receiving intravenous dicloxacillin [3,4,7,12]. Clinicians should be aware of hypokalemia as a complication of nafcillin use in view of the broad use of this antibiotic in the clinical setting.

## Figures and Tables

**Figure 1 antibiotics-07-00108-f001:**
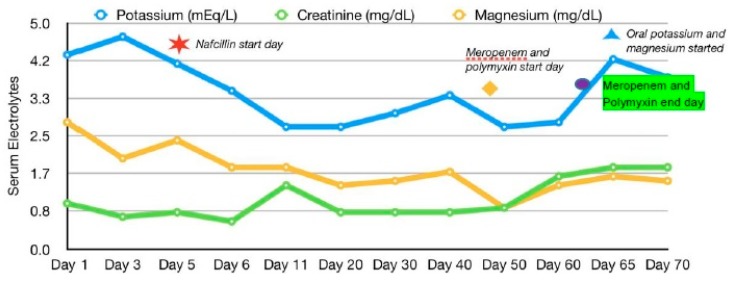
Electrolyte trends.

**Table 1 antibiotics-07-00108-t001:** Serum electrolytes during the hospital course.

Serum Electrolyte	Hospital Day														
	#1	#3	#5 ^%^	#6	#11	#20	#25	#30	#40	#45 ^@^	#50	#55	#60 ^∑^	#65 ^π^	#70
Sodium (mEq/L)	138	142	148	142	136	139	137	135	134	133	134	136	132	139	137
Potassium (mEq/L)	4.3	4.7	4.1	3.5	2.7	2.7	2.9	3.0	3.4	3.0	2.7	2.9	2.8	4.2	3.8
Chloride (mEq/L)	98	103	106	98	95	99	96	95	90	93	95	96	89	98	98
Bicarbonate (mEq/L)	22	26	27	29	27	28	29	26	27	24	26	26	28	25	29
Blood Urea Nitrogen (mg/dL)	40	30	32	19	20	6	6	5	5	7	9	5	9	9	11
Creatinine (mg/dL)	1.0	0.7	0.8	0.6	1.4	0.8	0.7	0.8	0.8	1.4	0.9	0.8	1.6	1.8	1.8
Magnesium (mg/dL)	2.8	-	2.4	1.8	1.8	1.4	1.7	1.5	1.7	1.6	0.9	1.3	1.4	1.6	1.5
Phosphorous (mg/dL)	4.2	-	3.6	2.9	-	2.4	3.0	3.1	3.1	2.2	2.1	2.4	2.3	2.2	2.6

Reference values (serum): sodium 135–145 mEq/L, potassium 3.5–5.0 mEq/L, chloride 98–108 mEq/L, bicarbonate 24–30 mEq/L, blood urea nitrogen 8–26 mg/dL, creatinine 0.5–1.5 mg/dL, magnesium 1.5–2.7 mg/dL, 2.5–4.5 mg/dL. Symbols: ^%^: nafcillin was started on day #5; ^@^: meropenem and polymyxin on day #2; ^∑^: meropenem and polymyxin were completed 3 days previously; ^π^: the oral potassium and oral magnesium start day.

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
