# Peer review of "A Case of Hypokalemia Possibly Induced by Nafcillin"

_antibiotics, 2018, doi:10.3390/antibiotics7040108_

Round 1

Reviewer 1 Report

Casano et al describe a case of nafcillin induced hypokalemia. This is something well described in the literature. The strength of the manuscript is in the evaluation of hypokalemia as well as the description of the mechanism of nafcillin induced hypokalemia 

There are a number of opportunities for improvement. 

The introduction does not discuss electrolyte abnormalities associated with antibiotic use. Since this is central to the case report it should be discussed in the introduction. 

Case presentation

There is a discrepancy between the presentation of R sided weakness ,  and the PE which shows b/l weakness. Some explanation of this is required perhaps correlating the imaging with the physical exam findings. Also the  ROS does not "show" weakness as this implies a physical exam finding but elicits this hx from the patient. 

How much ibuprofen was the patient taking for his pain prior to admission and was this continued in the hospital as this is another drug that can cause/predispose to hypokalemia. 

The major challenge of this paper is in the antibiotics, hypokalemia, and potassium replacement. Based on Table 2, the potassium is already dropping when the nafcillin is started. Additionally meropenem is well known to also cause hypokalemia and this should be addressed. It is likely the nafcillin is contributing to the hypokalemia throughout but when other antibiotics are involved it is more difficult, Also the day the oral potassium is started the potassium is normal. This is 3 days after meropenem was stopped consistent with it being a contributing factor. Based on the explanation for hypokalemia it is not clear why oral supplementation would be superior to IV and whether the oral supplementation truly helped

The discussion should acknowledge meropenem as a cause of hypokalemia. The discussion also states baseline K levels determine risk for hypokalemia. In reading reference 3, this patients baseline levels on admission did not increase the risk for hypokalemia. The details of the case report should be discussed in the context of what is known about risk factors for hypokalemia.

Author Response

POINT 1

The introduction does not discuss electrolyte abnormalities associated with antibiotic use. Since this is central to the case report it should be discussed in the introduction. 

RESPONSE 1

We added a discussion on electrolyte abnormalities in the Introduction part.

POINT 2

There is a discrepancy between the presentation of R sided weakness ,  and the PE which shows b/l weakness. Some explanation of this is required perhaps correlating the imaging with the physical exam findings. Also the  ROS does not "show" weakness as this implies a physical exam finding but elicits this hx from the patient. 

RESPONSE 2

Patient presented with right sided weakness which is the patient's intial complaint. On physical exam he was found to have decreased motor strength (right leg weaker than the left). MRI cervical spine was showing severe central stenosis of cervical spinal cord which can explain all his symptoms.

POINT 3

How much ibuprofen was the patient taking for his pain prior to admission and was this continued in the hospital as this is another drug that can cause/predispose to hypokalemia.

RESPONSE 3 

Patient stated he was taking only 1-2 pills every week.

POINT 4

The major challenge of this paper is in the antibiotics, hypokalemia, and potassium replacement. Based on Table 2, the potassium is already dropping when the nafcillin is started.

RESPONSE 4

Patient's baseline potassium level was 4.3 meq/l but after nafcillin was started his potassium levels started to trend down.

POINT 5

Additionally meropenem is well known to also cause hypokalemia and this should be addressed. It is likely the nafcillin is contributing to the hypokalemia throughout but when other antibiotics are involved it is more difficult, Also the day the oral potassium is started the potassium is normal. This is 3 days after meropenem was stopped consistent with it being a contributing factor. Based on the explanation for hypokalemia it is not clear why oral supplementation would be superior to IV and whether the oral supplementation truly helped.

The discussion should acknowledge meropenem as a cause of hypokalemia.

RESPONSE 5

Patient's potassium levels were already low even before meropenem which is clearly evident in the table and graph shown in the manuscript. We added a comment on meropenem as well.

Patient was getting intravenous potassium replacement on as-needed basis daily based on the potassium levels everyday. But once it was identified that nafcillin was the possible cause of hypokalemia, patient was started on oral potassium supplements daily based on how much he needed before. That is the reason why his levels normalized once oral supplements were started. We did not mean that oral supplements are superior.

POINT 6

The discussion also states baseline K levels determine risk for hypokalemia. In reading reference 3, this patients baseline levels on admission did not increase the risk for hypokalemia. The details of the case report should be discussed in the context of what is known about risk factors for hypokalemia.

RESPONSE 6

Patients with a higher baseline potassium level had a lesser likelihood of developing severe hypokalemia but a higher likelihood of having an acute drop in the potassium level. Although this patient’s baseline potassium was around 4.3 meq/l, there was an acute drop in his potassium levels to less than 3.5 meq/l once nafcillin was started.

Reviewer 2 Report

Fernando Casado and colleagues reported on a case of hypokalemia possibly related to the administration of nafcillin. In my opinion, the case is interesting and within the scopus of the journal. Below are major and minor comments on the technical merits of the paper.

Major comments

1) In my opinion, it would be preferable to change the title from “A case of nafcillin induced hypokalemia” to “A case of hypokalemia possibly induced by nafcillin”. Indeed, although attributing responsibility to nafcillin is theoretically reasonable, there is no definitive proof. Please note that this also applies to the entire text of the article, that should similarly indicate possibility rather than certainty.  

2) The text should be integrated by reporting the presence/absence of concomitant medications/conditions possibly responsible for hypokalemia (e.g., diuretics, corticoids, laxatives, diarrhea)

3) Were there also ECG abnormalities?

Minor comments

1) Antibiotics should be in lowercase (e.g., nafcillin instead of Nafcillin)

2) Correct form is methicillin-susceptible Staphylococcus aureus instead of Methicillin Sensitive Staphylococcus aureus

3) Another recent case of hypokalemia possibly induced by nafcillin has been reported that could be cited (Rukma P, et al. Am J Ther. 2018)

4) Did the patient signed an informed consent for publishing this case report? This information should be reported in line with ethical standards. 

Author Response

Major comments

POINT 1-

In my opinion, it would be preferable to change the title from “A case of nafcillin induced hypokalemia” to “A case of hypokalemia possibly induced by nafcillin”. Indeed, although attributing responsibility to nafcillin is theoretically reasonable, there is no definitive proof. Please note that this also applies to the entire text of the article, that should similarly indicate possibility rather than certainty.  

RESPONSE 1

we changed it to “A case of hypokalemia possibly induced by nafcillin”.

POINT 2-

The text should be integrated by reporting the presence/absence of concomitant medications/conditions possibly responsible for hypokalemia (e.g., diuretics, corticoids, laxatives, diarrhea)

RESPONSE 2

We added a comment on other medications. Patient stated that he had not been taking any other medications. He denied any diarrhea.

POINT 3

Were there also ECG abnormalities?

ECG was normal. There were no hypokalemic changes.

Minor comments

1) Antibiotics should be in lowercase (e.g., nafcillin instead of Nafcillin)

RESPONSE 1-

We edited it to lowercase

2) Correct form is methicillin-susceptible Staphylococcus aureus instead of Methicillin Sensitive Staphylococcus aureus

RESPONSE 2-

We edited it to methicillin-susceptible Staphylococcus aureus

3) Another recent case of hypokalemia possibly induced by nafcillin has been reported that could be cited (Rukma P, et al. Am J Ther. 2018)

RESPONSE 3-

We added this reference. Thank you

4) Did the patient signed an informed consent for publishing this case report? This information should be reported in line with ethical standards. 

RESPONSE 4-

We got the signed informed consent from the patient.

Round 2

Reviewer 1 Report

The manuscript describes a case of hypokalemia likely related to nafcillin use. The manuscript has been improved and the case has been put in the context of recent literature. 

There are some additional opportunities to further strengthen the manuscript.

It remains unclear why the IV potassium did not correct the hypokalemia as well as why there was a decision to switch to oral supplementation  and how that helped maintain K levels while the IV did not. 

The authors mention that there are reports of using a K sparing diuretic for antibiotic assoc hypokalemia. Was that done in this case?

The figure would benefit from noting where all antibiotics were stopped in addition to started directly on the figure. This will help the reader understand the temporal relationship between the difference antibiotics used and the potassium levels.  

Author Response

POINT 1

It remains unclear why the IV potassium did not correct the hypokalemia as well as why there was a decision to switch to oral supplementation  and how that helped maintain K levels while the IV did not. 

RESPONSE 1

IV potassium was given based on his daily potassium level in the morning and there were many instances when patient refused IV replacement because it was burning his arm.

We had to switch to oral supplementation because patient was being discharged from the hospital and it was not feasible for the nursing home to give IV replacement everyday. The patient was getting oral supplements 2 times a day but the IV replacements were on as-needed basis based on the potassium levels. When we checked the potassium levels in the evening on some days while the patient was getting was recieving IV replacement were normal, so IV replacement was not given during the evening and probably that was the reason why the morning potassium levels were low. But when the patient was getting oral potassium supplements he recieved it 2 times a day irrespective of the evening potassium levels. So, the morning potassium levels were normal while the patient was on oral potassium supplements.

POINT 2

The authors mention that there are reports of using a K sparing diuretic for antibiotic assoc hypokalemia. Was that done in this case?

RESPONSE 2

No. K sparing diuretic was not used in this patient. We decided to start only if he reamined hypokalemic despite being on supplements but his hypokalemia improved subsequently with potassium supplements.

POINT 3

The figure would benefit from noting where all antibiotics were stopped in addition to started directly on the figure. This will help the reader understand the temporal relationship between the difference antibiotics used and the potassium levels.  

RESPONSE 3

We have added the Meropenem and Polymyxin end day on the figure.

The Nafcillin end day was not added on the figure because patient was discharged on it and he recieved it in the nursing home and our figure only depicts hospital course.